# Prospective longitudinal study of tobacco company adaptation to standardised packaging in the UK: identifying circumventions and closing loopholes

Karen A Evans-Reeves, Rosemary Hiscock, Kathrin Lauber, Anna B Gilmore

Tobacco Control Research Group, Department for Health, University of Bath, Bath, UK

**Correspondence to**
Dr Rosemary Hiscock;
r.hiscock@bath.ac.uk

## ABSTRACT

**Objectives** UK standardised packaging legislation was introduced alongside pack size and product descriptor restrictions of the European Union Tobacco Products Directive to end tobacco marketing and misinformation via the pack. This paper aims to assess compliance with the restrictions and identify attempts to continue to market tobacco products and perpetuate misperceptions of harm post legislation.

**Design, setting and intervention** A prospective study of the introduction of standardised packaging of tobacco products to the UK.

**Participants and outcomes** We analysed commercial sales data to assess whether the legally required changes in pack branding, size and name were implemented. To explore any adaptations to products and packaging we analysed sales data, monthly pack purchases of factory-made (FM) cigarettes and roll-your-own (RYO) tobacco, tobacco advertisements from retail trade magazines and articles on tobacco from commercial literature (retail trade, market analyst and tobacco company publications).

**Results** One month after full implementation of the UK and European Union policies, 97% FM and 98% RYO was sold in compliant packaging. Nevertheless, tobacco companies made adaptations to tobacco products which enabled continued brand differentiation after the legislation came into force. For example, flavour names previously associated with low tar were systematically changed to colour names arguably facilitating continued misperceptions about the relative harms of products. Tobacco companies used the 1-year sell-through to their advantage by communicating brand name changes and providing financial incentives for retailers to buy large volumes of branded packs. In addition, tobacco companies continued to market their products to retailers and customers by innovating exemptions to the legislation, namely, filters, packaging edges, seals, multipack outers, RYO accessories, cigars and pipe tobacco.

**Conclusions** Tobacco companies adapted to packaging restrictions by innovating their tobacco products and marketing activities. These findings should enable policy makers globally to close loopholes and increase the potential efficacy of standardised packaging policies.

### Strengths and limitations of this study

► The strength of this paper lies both in the detail and depth of each analysis including our systematic analysis of retail-press advertisements (n=195) and commercial literature articles and reports (n=396) alongside pack purchases of eight top-selling brands and detailed sales information from Nielsen.

► Using multiple data sources has enabled findings to be verified by more than one source and enabled a greater understanding of the tobacco industry's motives for changes and innovations to their products and packaging. This insight would not have been possible using one dataset in isolation.

► By using the commercial literature, we were able to plug gaps in our knowledge, for example, we did not have Nielsen data on cigars or pipe tobacco but information in the retail and commercial literature revealed that these products were targets for innovation.

► Nielsen data did not tell us whether roll-your-own (RYO) packs were in branded or standardised packaging. Given that a minimum of RYO pack size of 30 g was a requirement of the Tobacco Products Directive we assumed that new 30 g RYO packs would switch to standardised packs at the same time.

► We do not know the extent to which Nielsen variant name changes lag behind and even reflect those printed on packs in retailers. However, we are reassured that the main name change patterns found in the Nielsen data were similar to those found in a previous UK convenience store study.

## BACKGROUND

Standardised tobacco packaging, one of the most significant policy threats to the tobacco industry in recent years, came into force in the UK on 20 May 2016. From this date all new factory-made (FM) cigarettes and roll-your-own (RYO) tobacco, manufactured or imported for UK consumption were to be sold in standardised packaging. Tobacco companies were given until 20 May 2017 to comply

**Box 1 Summary of restrictions UK standardised packaging legislation and the European Union Tobacco Products Directive (EU TPD) revision (implemented 20 May 2016–20 May 2017)**

**EU TPD:**

► Packaging opening: a unit packet of cigarettes may consist of carton or soft material and shall not have an opening that can be reclosed or resealed after it is first opened, other than the flip-top lid and shoulder box with a hinged lid. For packets with a flip-top lid and hinged lid, the lid shall be hinged only at the back of the unit packet.

► Pack size: packs must contain a minimum of 20 cigarettes and roll-your-own (RYO) tobacco must contain at least 30 g of tobacco.

► Impression creation: there must be no packaging elements that create an erroneous impression about the characteristics, health effects, risks or emissions of tobacco.

► Tobacco ingredients: no description of nicotine, tar or carbon monoxide content of a tobacco product.

► Flavour description: no reference to taste, smell or any flavourings or additives or the absence of any such thing.

► Flavourings (May 2020): no flavourings in any components such as filters, papers, packages, capsules or any technical features allowing modification of the smell or taste of tobacco products.

► Environment: no suggestion that a particular tobacco product has improved biodegradability or other environmental advantages.

► Resemblance: no resemblance to a food or cosmetic product.

► Promotion: no suggestion of economic advantage of one product over another—no price marked packs.

**UK standardised packaging:**

► Pack external colour: external packaging of cigarettes or RYO tobacco is Pantone 448C (matt finish).

► Pack internal colour: internal colour of cigarette or RYO packaging must be white or Pantone 448C (matt finish).

► Product name appearance: Brand name should appear on one line in Helvetica unweighted typeface no larger than point 14 and brand variant name should follow the same rules but in no larger than 10 point. The start of each word can be an upper case letter but the rest of the word must be lower case.

► Noise and smell: packaging must not make a noise or produce a smell that is not normally associated with the packaging.

► Packaging evolution: the packaging must not change after retail sale, for example, heat activated inks, embellishments designed to appear gradually over time, scratch panels and so on.

► Packaging shape: cigarette packets must be made of carton or soft material and be cuboid in shape and RYO may be in a cuboid box, a cylinder or a pouch (bevelled or rounded edges are permitted).

► Stick design: cigarette stick paper must be plain white (matt finish), the filter must only be coloured in such a way as to imitate cork. The stick may have the brand name and variant printed in black Helvetica type no larger than eight point in normal weighted regular typeface not more than 38 mm from the filter end of the cigarette. The start of each word can be an upper-case letter but the rest of the word must be lower-case.

with the law; a period referred to as the 'sell-through'. During this time, companies were not permitted to print any new branded packets.[1 2] The UK legislation[1] was implemented in concert with the revision of the 2001 European Union (EU) Tobacco Products Directive (TPD)[3] which placed further restrictions on packaging and naming of brands (box 1).[3]

After Australia became the first country to introduce standardised packaging in December 2012, tobacco companies responded with more evocative and descriptive tobacco product names, including colours to represent the previous pack colour, thereby continuing the connotations associated with these colours.[4 5] Simultaneously companies reduced the total number of brands sold and renewed their focus on value for money brands by increasing the number and length of cigarettes in a pack and introducing menthol variants in this price range.[6] In response to standardised packaging in New Zealand and other markets there has been a significant increase in flavour capsule variants (FCVs) across all market price points,[7] which may threaten standardised packaging's effect on deterring smoking initiation given that these products appeal more to non-smokers and non-daily smokers than to daily smokers.[8 9]

While the UK and EU legislation combined closed some of loopholes in the Australian legislation, by prohibiting product names that create an erroneous impression about the health effects and requiring minimum pack sizes for both cigarettes and RYO,[10] concerns remain. The current legislation permits the use of colour descriptors, bevelled edges on packs, cigarette filter technology innovations and the TPD ban on menthol flavouring (including capsule technology) does not come into force until May 2020. Furthermore, cigars, cigarillos and pipe tobacco and RYO filter tips are exempt from the legislation and are still sold in branded packaging, while wholesaler multipacks are still allowed branded 'outer' wraps.

Recently published research suggests that, prior to the full implementation of standardised packaging in the UK, tobacco companies added smaller pack sizes such as 17 sticks to their brand variant portfolios,[11] introduced colour descriptors to brand family and brand variant names and made product changes, particularly to FM cigarette filters.[12] Examples of name change patterns associated with UK standardised packaging restrictions have been described qualitatively[10 12 13] but not quantitatively. The scale of these name changes and if/how these changes were communicated to the public is not clear from published research. Previous work has also only explored up to 2 months postlegislation.

Therefore, the purpose of this paper was to use mixed methods to combine data from four different data sources up to 7 months after full-implementation to systematically and rigorously examine how compliant tobacco companies were with the legislation, and to explore any attempts they made to circumvent it. This including any actions to reduce the immediate impact of the legislation during the sell-through. We assessed how the sell-through period was used, how and if tobacco company marketing adapted including any changes to products or packaging that would endure after the legislation was fully implemented, and whether and how products were marketed to retailers and consumers. The findings of this paper

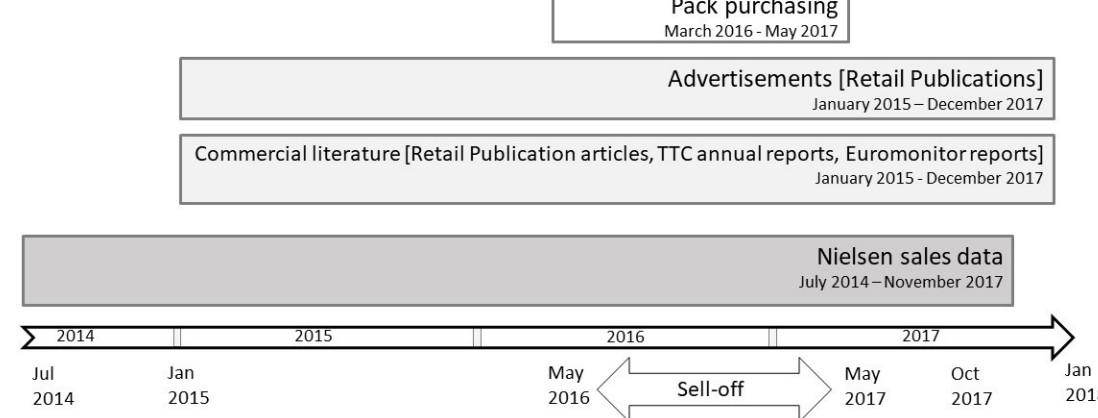

**Figure 1** Timeline of data collection by data source. TTC, transnational tobacco company.

have global significance as they can help close loopholes during the design and implementation of standardised packaging legislation in other jurisdictions.

## METHODS
### Data sources
We utilised four data sources: (1) purchases of eight top-selling tobacco brands; (2) tobacco advertisements in popular retail trade publications; (3) other commercial literature: retail press articles and advertorials, tobacco company annual reports and Euromonitor market analyst reports; and (4) Nielsen data (figure 1). While compliance with the legislation was primarily assessed using the Nielsen data, the evidence collected from each data source was utilised to assess whether, and how, tobacco companies adapted to the legislation and continued to market their products. We triangulated data from these different datasets, to verify the existence and purpose of adaptations.

### Pack purchasing and observation
We conducted monthly pack purchases between March 2016 and May 2017 to assess visual and sensory changes over time in the top-selling FM and RYO products (table 1) for different price segments. Price segments were defined from commercial literature and Nielsen data tobacco price data of sales between 2008 and early 2016.[11] We purchased packs from five different FM price segments and two different RYO price segments. We also included British American Tobacco's (BAT's) top selling brand in the UK as an addition as BAT did not have a top-selling brand in any of the identified price-segments and we wished to include all four Transnational Tobacco Companies (TTCs) in the observational study.

Additionally, we observed price lists of cigar and pipe tobacco brands available in-store and the appearance of 'outers' (the cover wrap used to bind multiple packs of RYO or FM together for distribution) where visible.

### Advertisements in the retail press
We searched hard copies of the most widely circulated paid for retail and wholesale publications *The Grocer, Wholesale News* and *Retail Newsagent* between January 2015 and December 2017 for all tobacco advertisements.

We coded 195 advertisements for the attributes marketed to retailers, for example, new brand variant, packaging, filter, limited edition, price, retailer profit. We examined the distribution of these marketing themes by price segment (online supplementary table S1). We separately assessed advertorials for relevance to standardised packaging and noted the main messages communicated.

### Commercial literature review
We searched high circulation online retail publications (*The Grocer, Wholesale News, betterretailing.com* (which includes *Retail Newsagent* and *Retail Express), talkingretail. com*), to identify articles relevant to tobacco packaging and marketing (January 2015–December 2017; table 2). We also included two reviews of 2017 published in 2018, Euromonitor reports on tobacco, and tobacco company annual reports covering this period. We used NVivo 10, to code articles for any evidence of circumvention of the legislation and used quotes taken from the commercial literature as examples of each circumvention (online supplementary table S2).

### Nielsen data
Nielsen collates data on tobacco sales from nearly 90% of UK supermarkets and a stratified sample of 15% of convenience stores.[11] For each product, Nielsen records hierarchically the tobacco brand, brand family, brand variant and then specific features of the pack denoted by a unique serial number known as a stock keeping unit (SKU) (eg, size, pricemarked).[11] In November 2017 there were 71 brands, 97 families, 241 variants and 1022 SKUs (table 3).

**Table 1** Pack purchase data of the top selling brand in each price segment

| Price segment (average price per pack/stick)* | Top selling products name and packsize | | |
| | March 2016 (2 months prior to the start of sell-through) | May 2017 (full-implementation) | Co |
| --- | --- | --- | --- |
| FM | | | |
| Premium (£6.76/£0.42) | Marlboro Gold (20) | Marlboro Gold (20) | PMI |
| Midprice† (£5.61/£0.36) | L&B Original Lambert & Butler Silver (20) Mayfair King Size (19) | Lambert & Butler Original Silver (20) Mayfair King Size (20) | IMT JTI |
| Value (£5.10/£0.33) | L&B Blue Lambert & Butler Real Blue (19) | L&B Blue Real Blue (20) | IMT |
| Subvalue (£4.68/£0.29) | Carlton Red (19) Rothmans of London‡ (17) | Carlton Red (20) Rothmans Blue (20) | IMT BAT |
| RYO§ | | | |
| Premium (£6.97/£0.17) | Golden Virginia The Original (25 g) | Golden Virginia The Original (30 g) | IMT |
| Midprice (£5.81/£0.15) | Amber Leaf (25 g) | Amber Leaf Original (30 g) | JTI |

*Weighted average price per pack/stick of all products sold within tobacco industry price segments as described by previous analysis (December 2015 prices—inflation adjusted to 2008 values). One RYO stick was estimated to be 0.5 g tobacco.[11 47]
†Midprice was originally two major segments: Lambert & Butler was the leading upper midprice product and Mayfair the leading lower midprice product.
‡None of the top selling brands in each price range was owned by BAT. For completeness we added Rothman's of London (a subvalue brand)—the top selling brand for BAT.
§No RYO value segment (average pack/stick price: £4.28/£0.14) product was purchased due to lower sales volumes even of the leading brand (Gold Leaf) in this segment.
BAT, British American Tobacco; FM, factory made; IMT, Imperial Tobacco; JTI, Japan Tobacco International; PMI, Philip Morris International; RYO, roll your own.

## Data analysis

### Compliance

Monthly Nielsen sales data were used to examine compliance with the removal of branding, pricemarking and small pack sizes (box 1), between July 2016 (when the first standardised packs first were sold)[13] and November 2017. For FM, Nielsen data identifies which packs are standardised and which are branded. For RYO they do not. However, as 30 g packs of RYO were negligible prior to the legislation RYO packs of 30 g or more were treated as standardised.

### Name changes

We identified the number, names and price segments of all brand variants whose name changed between July 2014 and November 2017. July 2014 was used as a starting point for this analysis as we wanted a time before standardised packaging was passed by the Government in January 2015. We coded and counted whether each name change was required for compliance with the legislation and recorded the type of name change, for example, colour added or adjective added.

### Innovation targeting

To understand whether innovations were targeted at particular price segments, we compared the distribution of name changes and new variants in November 2017 with the distribution of price segments prior to the decision to implement legislation in July 2014. Differences in the number of variants were tested using Chi-Squared ($\chi^2$) tests and Fishers exact tests (when the expected number of cases was less than five). To achieve sufficient numbers for analysis, price segments were merged into four categories[1]: FM premium and midprice,[2] FM value and subvalue,[3] RYO premium and midprice,[4] RYO value.

### Patient and public involvement

KAE-R and RH discussed standardised packaging policy with the UK Centre for Tobacco and Alcohol Studies University of Nottingham panel of smokers and former smokers (now known as the Tobacco and Nicotine Discussion Group) in October 2017. The group were asked to discuss their thoughts on and experiences of the policy and whether they had noticed any changes to tobacco products and packs both in the sell-off period and thereafter. These discussions helped inform our research questions but were not included in our analysis.

## RESULTS

Four main strategies were identified from multiple data sources. We present the evidence for each in turn.

**Table 2** Commercial literature review January 2015 to December 2017

| Source | Search terms (depends on search procedure of website) | Articles/reports found | Articles/reports included in final analysis |
|---|---|---|---|
| Retailer magazines | | | |
| Convenience and independents | | | |
| Betterretailing.com: Retail Newsagent Retail Express | 'tobacco' | 350 | 181 |
| TalkingRetail.com | 'tobacco' | 54 | 49 |
| Wholesalers and FMCG | | | |
| Wholesale News | 'tobacco' or 'cigarette' or 'ryo' or 'roll +your+own' or 'hand+rolled' or 'cigar' or 'cigarillo' | 144 | 30 |
| The Grocer | 'tobacco' or 'cigarette' or 'ryo' or 'roll+your+own' or 'hand +rolled' or 'cigar' or 'cigarillo' | 820 | 104 |
| Industry analyst | | | |
| Euromonitor Passport | 'tobacco' | | 20 |
| Tobacco company | | | |
| Annual reports from the big four tTTCs: JTI, IMT, BAT and PMI | | 12 | 12 |
| **Total** | | 1380 | 396 |

BAT, British American Tobacco; IMT, Imperial Tobacco; JTI, Japan Tobacco International; PMI, Philip Morris International.

### Strategy 1: keep branded packets on the market as long as possible

Six months into the year-long sell-through period, 96% of FM and 82% RYO were still sold in branded packs (figure 2 and online supplementary table S3). Most FM (70%) and RYO (65%) packs switched to standardised packs between January and April 2017. One month after the full-implementation of the UK and EU policies, 97% of cigarettes and 98% of RYO were sold in compliant packaging. By November 2017, non-compliant sales declined to 1% of FM and negligible volumes of RYO.

The commercial literature suggests that tobacco companies produced greater volumes of their branded FM and RYO tobacco prior to the May 2016 manufacturing deadline to keep branded packaging on the market for as long as possible during the sell-through.[14] Both the Nielsen data[11] and the trade press (advertisements and commercial literature) revealed an increase in small pack sizes ahead of the sell-through period. Tobacco companies offered incentives and promotions on branded packs to retailers to encourage sales and loyalty to their products, examples included cash rewards and loyalty points

**Table 3** Hierarchical brand architecture available from the Nielsen data

| Hierarchy of Nielsen data | Example | N (UK market) | |
|---|---|---|---|
| | | July 2014 | November 2017 |
| Brand | Marlboro | 86 | 71 |
| Brand family Products with the same brand name but sold at different price points | Marlboro <u>Bright Leaf</u> | 114 | 97 |
| Brand variant Product at same price point but with different length (eg, superkings, kingsize), flavour or other characteristics | Marlboro Bright Leaf <u>Platinum</u> | 282 | 241 |
| Stock keeping unit Each individual barcoded product including specific pack characteristics: pack size, whether the pack is price-marked, and, for FM but not RYO, whether standardised or branded | Marlboro Bright Leaf Platinum <u>10 s multipack not price marked</u> | 930 | 1022 |

FM, factory made; RYO, roll your own.

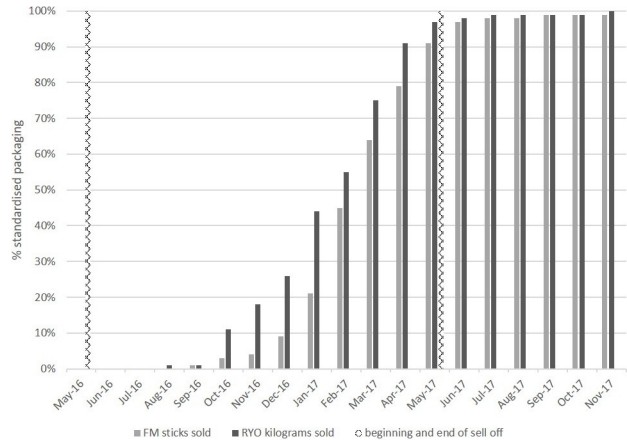

**Figure 2** Proportion of FM sticks and RYO* (by weight) sold in standardised packaging over time. FM, factory made; RYO, roll your own.

for selling particular brands. By early 2017 manufacturers were described as being in a price-war.[15 16] To allay retailers' concerns about being left with non-compliant stock after the deadline, tobacco companies offered to buy them back (online supplementary table S2; Strategy 1.3).[17]

Overall, tobacco companies decreased the number of brands, brand families and brand variants between July 2014 and November 2017 as they were consolidating their portfolios and focusing on the brands which could offer growth. The increase in SKU between 2014 and 2017 reflects (1) the pre-legislation increase in small pack sizes and innovations such as flavours and capsules; (2) the addition of standardised packs FM SKU and 30 g RYO SKU; (3) that many SKU in branded packs remained on the market in November 2017 but selling only tiny volumes.

### Strategy 2: maintaining brand variant differentiation through name changes

One-third (35%; 123/353) of variants changed name between July 2014 and November 2017. Less than half of these changed name to comply with legislation (42%, n=52/123) and most of these removed flavour names as the legislation required but substituted with other names, most often a colour (table 4). Most notably, menthol changed to green, FM smooth became bright or sky blueand RYO smooth changed to yellow.

Of the variants which changed name when it was not necessary for legislative compliance (n=71/123, 58%), colours were introduced for the first time as a substitute for the previous pack colour, for example, *Marlboro* with the red chevron on the pack became Marlboro Red. Similarly, to distinguish brand variants from one another and to continue previous brand variant identities, other adjectives were added to brand names, most notably adjectives accompanied a colour, for example, *real* blue, *bright* silver, *legendary* black. In addition, during the study period, new variants appeared in the Nielsen dataset with names which included elements that were novel: 'signature',

'ome' and 'silver strand' in May 2015, 'Eagle', 'Black Russian' and 'Colours" in November 2015 and 'Chill' in May 2016 . Additionally, two new Golden Virginia RYO variants appeared on the market in November 2015 named 'Midnight' and 'Sunrise'. Commercial literature revealed that these indicated rich and smooth flavours, respectively.[18] The first brand variant in a brand family sometimes had 'original' added, for example, Amber Leaf RYO became Amber Leaf original.

In October 2016, 5 months into the sell-through period and before Imperial Tobacco (IMT) had made any of its legally required name changes to its products, the company outlined its brand renaming strategy in an advertorial in the retail press.[19] The advertorial explained that all products traditionally presented as full flavour would become 'real' or 'original' and be associated primarily with the colour red. Those previously denoted as 'smooth' would change to 'bright' and become associated with the colour blue. Menthol would become green and capsule would become Crushball (online supplementary table S2: Strategy 2). IMT also provided retailers with materials to help explain changes to consumers.[19] Some companies also advertised name changes directly to consumers through pack inserts (eg, BAT's Rothmans of London included a pack insert to inform its smokers that it was to become Rothman's Blue).

### Strategy 3: focus on lower price segments

During the study period, 68% of new variants recorded by Nielsen were FM and 32% RYO. Given that in July 2014, 81% variants on the UK market were FM and 19% were RYO (table 5), more new RYO variants were introduced than expected (p=0.018).

Half (50%) of all new variants were introduced in the FM value and subvalue price segments, 22% FM premium and midprice, 15% RYO premium and midprice and 13% RYO value. However, prior to the decision to implement standardised packaging (July 2014), the distribution of the market was 50% FM premium and midprice, 35% FM value and subvalue, 8% RYO premium and midprice and 7% RYO value. Thus, new variants were under-represented among FM premium and midprice brands (p=0.001).

Similarly, compared with July 2014, name changes in the study period were over-represented among FM value and subvalue (p=0.048) and under-represented among FM premium and midprice brands (p=0.002).

Companies introduced more menthol and flavoured capsule variants (FCVs) to the lower price segments which traditionally offered fewer of these products compared with premium and midprice segments. Since July 2014, 13 menthol or flavoured capsule variants were introduced to the value and subvalue price segments compared with just two new brand variants in the premium and midprice range (online supplementary table S4). Overall, the number of brand families and brand variants declined. Tobacco companies reported concentrating on fewer brand families so that they could improve the 'quality of

**Table 4** Nielsen data FM and RYO brand variant name changes from July 2014 to November 2017

| | | N | Additions/substitutions |
|---|---|---|---|
| **Legally required actions** | | | |
| Taste removed | Replacement | | |
| Menthol | Colour | 20 | Green |
| | No replacement | 2 | |
| Smooth | Colour+adjective | 12 | Bright blue/yellow, sky blue |
| | Adjective | 5 | Bright, sky |
| | Colour | 3 | Blue, yellow |
| | Other Substitution | 1 | Fine |
| Other taste ('fresh burst', 'fresh taste', 'ice capsule', 'subtle flavour') | Other Substitution | 3 | Crushball, dual |
| | No replacement | 1 | |
| Environmental advantages removed | | | |
| 'Natural' | Other Substitution | 2 | Blue, king size |
| | No replacement | 3 | |
| Total | | 52 | |
| **Discretionary actions only** | | | |
| Colour+adjective added | | 9 | Original silver/blue/black, real blue/red |
| Adjective added | | 8 | Bright, real, legendary |
| Colour added | | 7 | Blue, red, black |
| 'Original' added | | 6 | |
| Size (king size, superkings) removed | | 6 | |
| Mixed and miscellaneous | | 35 | Mixed: for example, colour added and size removed Miscellaneous: for example, location added or removed - Rothman's of London became Rothman's Blue |
| Total | | 71 | |
| Grand total | | **123** | |

FM, factory made; RYO, roll your own.

growth' and 'cut the level of complexity and cost in the business'.[20]

Nearly two-thirds of the advertisements captured from the retail press, 124/195 (64%) advertised value or subvalue brands. Similarly, for new products, the majority of advertisements (30/38; 79%) were for value or subvalue brands. Only one advert was a new premium offering.

The commercial literature found tobacco companies innovating within lower price segments, introducing packs with fewer cigarettes prior to the sell-through[21–23] followed by a price-war in the lower priced segments with some manufacturers reportedly selling at a loss to gain market share.[15] By the end of 2017 tobacco companies stated that smokers expected better quality products in lower price segments after standardised packaging and Minimum Excise Tax legislation raised the price of the cheaper brands and so 'premium features' in lower priced cigarettes appeared (online supplementary table S2: Strategy 3).[22 24] For example, a redesign to PMI's Chesterfield (subvalue) was advertised to retailers in December 2017, citing a new 'firm' filter and, as with premium brand Marlboro, a new bevelled edge box.[23 25]

### Strategy 4: innovating exemptions to the legislation

The commercial literature and pack purchase study suggested that tobacco companies continued to innovate and market their tobacco products by focusing on exemptions to the legislation (online supplementary table S2: Strategy 4).

#### Pack modifications that endured postlegislation

During the sell-through, the pack purchase and commercial literature review revealed that FM Marlboro 10 s and RYO *Amber Leaf*, *Golden Virginia* and *Cutter's Choice* were sold in reusable tins, arguably allowing consumers to decant cigarettes bought postlegislation into branded tins. Selling RYO in tins is not novel but cigarettes have rarely been sold in tins previously.

**Table 5** Nielsen data. Price segmentation of new variants and variant names changes between July 2014 and November 2017 compared with July 2014 baseline

| | Baseline distribution of variants (July 2014) | | Name changes (July 2014–November 2017) | | | New variants (July 2014–November 2017) | | |
|---|---|---|---|---|---|---|---|---|
| | N | % | N | % | P value* | N | % | P value* |
| Tobacco type | | | | | | | | |
| FM | 225 | 81 | 125 | 79 | 0.527 | 48 | **68** | 0.018 |
| RYO | 54 | 19 | 35 | 21 | | 23 | **32** | |
| Price segment | | | | | | | | |
| FM premium and midprice | 103 | 50 | 48 | **33** | 0.002 | 10 | **22** | 0.001 |
| FM value and subvalue | 72 | 35 | 66 | **45** | 0.048 | 23 | **50** | 0.054 |
| RYO premium and midprice | 17 | 8 | 20 | **14** | 0.097 | 7 | 15 | 0.164† |
| RYO value | 15 | 7 | 12 | 8 | 0.735 | 6 | 13 | 0.602† |

*$\chi^2$ comparing with July 2014 distribution.
†Fishers exact test comparing with July 2014 distribution.
FM, factory made; RYO, roll your own.

Pack purchases revealed that best-selling premium brand family *Marlboro* changed significantly in July 2016 (2 months into sell-through), with the introduction of bevelled pack edges and a *pro-seal* closing mechanism (figure 3). Both features fundamentally changed the tactile nature of the pack which endured after full-implementation in May 2017 (online supplementary file 2 for a video of the modified Marlboro pack). To be compliant with the legislation, these packs must have been printed prior to May 2016.

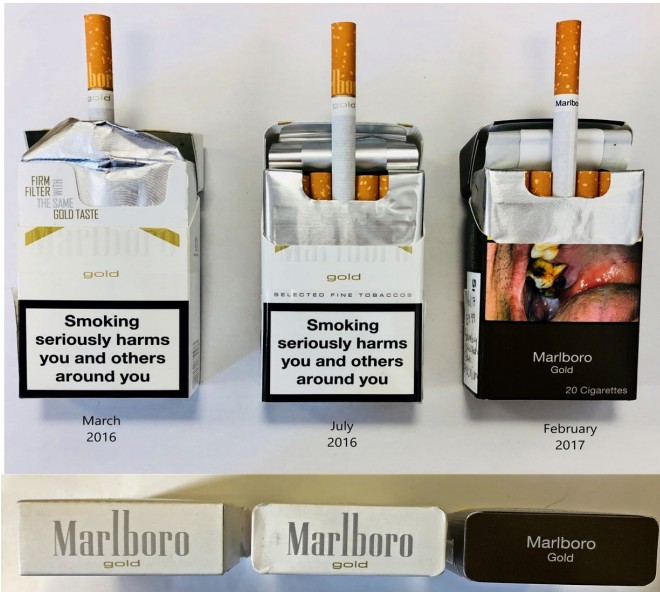

**Figure 3** The evolution of Marlboro gold packaging from a straight edge pack to a branded bevelled edge pack with a new internal packet with pro-seal sealing mechanism, to standardised packaging maintaining these innovations.

### Extra sticks

Given that there is no restriction on the maximum number of sticks per pack, packs with more than 20 continued postlegislation. During the sell-through (May 2016–May 2017) sales of packs with 23 and 24 sticks increased from 7 to 18 million sticks. Sales peaked at 25 million sticks in August 2017 and declined to 21 million sticks in November 2017.

### Branded outers

The retail literature and in person observations revealed that multipack wholesale outers for FM and RYO are branded (figure 4). Outers can be seen by customers during tobacco product gantry restocking. Advertisements in the retail press showed images of branded tobacco products including branded outers postlegislation.

### RYO accessories

Increased innovation was observed among RYO accessories post legislation with retail press articles in November 2017 and February 2018 referring to '"ultra slim" and slim filters and papers, biodegradable filters, and 'menthol tips' (online supplementary table S2; Strategy 4.4).[24 26]

### Cigars

Cigars, which can be sold in branded packs, single sticks without pictorial health warnings, at a relatively low price and with good profit margin, were identified as an opportunity for growth.[18 23 27–29] Euromonitor's models suggested growth in cigars and cigarillos sales volumes (390–494 million units) and value (£292–£348 million) from 2015 to 2018 (online supplementary table S5). Towards the end of 2017, JTI and cigar company Ritmeester were holding social events to build relationships with retailers (online supplementary table S2; Strategy 4.5).[23]

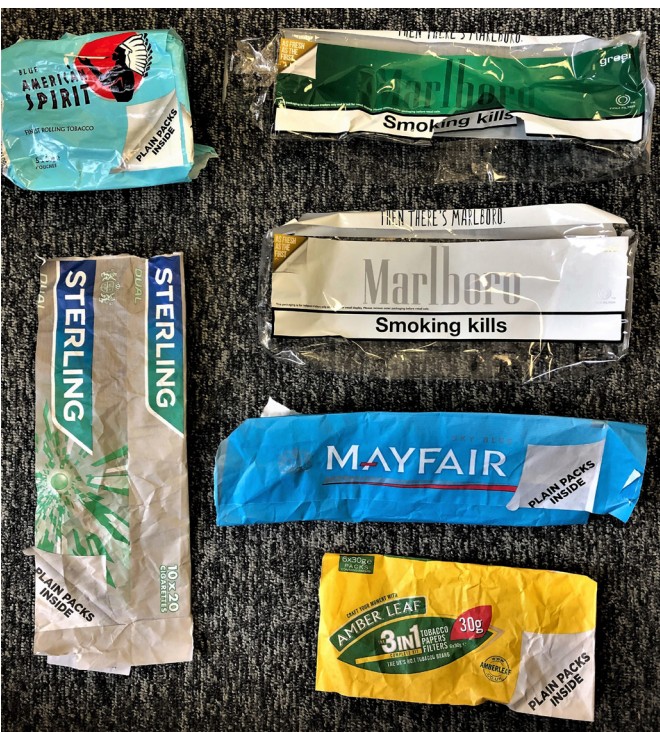

**Figure 4** Branded outers post May 2016 (full-implementation).

## Pipe tobacco

The retail literature revealed that one small tobacco manufacturer, *Gawith Hoggarth*, deliberately marketed pipe tobacco as RYO to circumvent the minimum pack size restrictions on RYO (online supplementary table S2; Strategy 4.6).[30]

## DISCUSSION

UK standardised packaging legislation and the EU TPD placed restrictions on tobacco packaging and marketing in the UK. Overall, although compliance with the restrictions (removal of branding, small pack sizes and non-compliant names) was not 100% 1 month after full implementation, the majority of non-compliance could be explained by less popular tobacco products that were not widely circulated. However, clarity is needed about the legality of isolated incidents such as the compliance of the *Marlboro* resealable pack and Gawith Hoggath's marketing of its *Kendal* pipe tobacco as RYO (online supplementary video 1).

Technical compliance aside, this paper identified four key strategies used by tobacco companies to circumvent the legislation.

First, the retail literature suggests that tobacco companies used the 12-month sell-through to their advantage, keeping branded stock on the market as long as possible, and using the time to communicate name changes and new brand variants to retailers and customers. Ahead of the sell-through companies increased production of branded packs, introduced smaller pack sizes to enable

more affordable offerings, and encouraged retailers to buy large quantities of branded stock at reduced prices. Other countries had a shorter sell-through period with 3 months for Australia and New Zealand[31 32] and 7 months for France.[33] Governments considering the policy in other countries should therefore consider mandating a short sell-through period.

Second, although compliant with legally required name removals, tobacco companies implemented a standard name change formula that maintained the brand identity and differentiation of the three-broad flavour categories[1] full flavour[2]; smooth[3]; menthol.[19] In 2002 terms such as 'mild' and 'light' were prohibited to curb misperceptions of relative harm. However, the continuation of the colours of the previous packs such as the gold and white pack for *Marlboro 'lights'* and the introduction of terms such as smooth for other brands sustained these misperceptions.[34 35] In line with the power of colour in brand identity,[35–37] this study shows that tobacco companies changed '*full-flavour*' variants to *red, original,* or *real; menthol* variants to *green*; and *smooth variants* to *bright* and *blue* (or *yellow* for RYO). As in 2002, the current restrictions on tobacco product name were designed disrupt misperceptions of relative harms based on flavour descriptions. However, by maintaining the broad flavour categories, through the use of colour descriptors, such misperceptions are likely to endure. Additionally, cigarette packs with filter descriptions such as 'advanced' and 'firm' filter are perceived as less harmful by existing smokers and therefore perpetuate the perceptions that some tobacco brand variants are less harmful than others.[38] Given that companies continue to innovate their product names and descriptions it may be considered necessary to follow the example of Uruguay where only one brand variant is permitted per brand and no new brands are allowed.[39]

Third, Nielsen data and the commercial literature revealed that tobacco companies were fighting fervently for market share in the cheaper price segments with a price war preimplementation. In the lower price segments, tobacco companies implemented more name changes, introduced more new brand variants (including menthol and flavour capsules) and more RYO variants and placed more advertisements for lower priced products then they did for premium and midprice products. Other work suggests that RYO is a lower cost alternative to smokers who may otherwise quit and so this may explain the introduction of more RYO offerings in lower price segments.[40–42] Similarly, menthol and particularly flavour-capsule variants appeal more to non-smokers and non-daily smokers compared with smokers and may therefore recruit non-smokers.[8] Flavoured cigarettes including capsule flavourings will be outlawed in the EU by May 2020. With the UK set to leave the EU in 2019, the tobacco industry may use this opportunity to try and roll back tobacco control regulations made under the EU TPD.

Our fourth finding that tobacco companies are innovating tobacco products, features and accessories not

covered by the UK or EU legislation for growth provides may provide a rationale for expanding the legislation to include products such as RYO filters and papers, standardised packaging for cigars, cigarillos, pipe tobacco, wholesale outer wraps, standardised pack edges, maximum pack sizes and prohibitions for innovations to pack seals. Tobacco companies, such as IMT in the UK have recently introduced a whole series of FCVs to their *Rizla* filter tip product offerings for RYO tobacco.

The strength of this paper lies both in the detail and depth of each analysis including our systematic analysis of retail press advertisements (n=195) and commercial literature articles and reports (n=396) alongside pack purchases of the top selling brands and detailed sales information from Nielsen. Using multiple data sources enabled findings to be verified by more than one source and enabled a greater understanding of the tobacco industry's motives for any changes made to their products and packaging. In addition to the evidence presented by others,[1 13 14] by following these data sources up to 7 months postlegislation we were able to observe tobacco companies increased focus on innovations to exemptions to the legislation that offered opportunities for growth, for example, RYO filters and accessories and cigars.

Nevertheless, despite evidence from an IMT whistleblower[43] and Philip Morris International's own words,[44] our data did not reveal price mark stickers on tobacco products. This may be because we bought our eight brands from a large supermarket and not a convenience store and because this industry strategy was not highlighted in the retail literature due to the questionable legality of this strategy.[45]

Resources prevented us from acquiring Nielsen data on cigars or pipe tobacco and from purchasing more than one brand per price segment in the pack purchasing study element. However, the retail literature alerted us that cigars and other products were targeted as growth opportunities and, although not able to fully capture the sensory nature of brands, our analysis of the advertisements ensured that we saw many, if not all, innovations being promoted to retailers.

Nielsen data do not record whether RYO products are sold in standardised packs. We assumed that 30 g packs were always in standardised packs and that larger packs would switch to standardised packs at the same time. It is possible that this did not occur. However, the temporal patterns of name changes and pack sizes in the Nielsen data were similar for FM and RYO and it is therefore reasonable to assume that branding was removed at the same time. Nielsen model prices and volumes for the UK-based on a census of the major supermarket sales and a rolling sample of convenience stores. Although, we do not know the extent to which Nielsen variant name changes lag behind and even reflect those printed on packs in retailers, the main name change patterns found in the Nielsen data were similar to those found in a UK convenience store study[46] and two evaluations of the introduction of standardised packaging in Australia.[4 5]

Given that the tobacco industry is attempting to circumvent standardised packaging legislation, other countries considering the policy should consider how to make regulations as comprehensive as possible to prevent the exploitation of continued marketing opportunities. In summary, the evidence in this paper suggests a number of possible policy options, namely that long sell-through periods should be avoided and that restricting tobacco products to one brand variant per brand may be the only comprehensive way to prevent misperceptions of harm and an that extending regulations to include other tobacco products, features and accessories should be considered.

**Acknowledgements** The authors would like to thank Dr Rob Branston who read and commented upon a near final draft and Ilhan Marsal who helped with early preparation of the advertisements and pack purchased data. We would also like to acknowledge Nielsen (UK) and Public Health England for provision of the sales data.

**Contributors** ABG designed the study with contributions from KAE-R and RH. KAE-R led the writing of the paper and designed, conducted and analysed the pack purchase element of the study. RH analysed the Nielsen data in addition to downloading and analysing the commercial literature. KL collated the retail literature advertisements, involving visits to the British Library and analysed the data. All authors read and approved the manuscript.

**Funding** This work was supported by Cancer Research UK grant number C27260/A23168. The authors are members of the UK Centre for Tobacco & Alcohol Studies, a UK Clinical Research Collaboration Public Health Research Centre of Excellence whose work issupported by funding from the Medical Research Council, British Heart Foundation, Cancer Research UK, Economic and Social Research Council, and the National Institute for HealthResearch under the auspices of the UK Clinical Research Collaboration (MR/K023195/1).

**Competing interests** None declared.

**Patient consent for publication** Obtained.

**Provenance and peer review** Not commissioned; externally peer reviewed.

**Data availability statement** Data are available in a public, open access repository. Data may be obtained from a third party and are not publicly available.

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
