## [Reviewer comments · BMJ Open]

ARTICLE DETAILS

TITLE (PROVISIONAL)	A prospective longitudinal study of tobacco company adaptation to standardised packaging in the UK: identifying circumventions and closing loopholes
AUTHORS	Evans-Reeves, Karen; Hiscock, Rosemary; Lauber, Kathrin; Gilmore, Anna

VERSION 1 – REVIEW

REVIEWER	Janet Hoek Departments of Public Health and Marketing, University of Otago, New Zealand I have met Anna Gilmore, who hosted me when I visited the University of Bath in 2015.
REVIEW RETURNED	16-Jan-2019

GENERAL COMMENTS	I enjoyed reading this thoughtful and very comprehensive study; it makes a new contribution by drawing together different data sources and triangulating the evidence from these. I have a few suggestions that I hope will be useful, and have outlined these below. General Comments I thought the MS could have drawn more on the international literature; the Scollo references from Australia are very important and well used, but there are also other studies documenting product innovation and how this accelerated as plain packaging was announced.[1] Earlier work also discusses how innovation could threaten plain packaging's effect on deterring initiation;[2, 3] outlines the effects of variant names [4, 5] and colours, [6] and overviews policy loopholes and how these might be addressed.[7] These are just a few of the papers I know about that could illustrate how UK evidence both reflects and is supported by international data. Given how comprehensive the data sources are, I hope you will see value in expanding the literature review. In places, the MS would benefit from some editing and careful proofreading. For example, the references have numerous errors and I'm not sure how easy it would be for someone not involved in the project to locate some items. EndNote does strange things to some references (see refs 1,2, 30, for example) and all the capitals seem to have disappeared (e.g. from ITC, UK etc.). Shortening sentences, reducing prepositional phrases, employing the active voice, and using more precise language could make the MS easier to read. Please also check your use of commas. Some specific suggestions are: Abstract
---

	1. Could you use a singly date for assessing compliance? I would have thought the best date was the actual date of transition (May 20, 2017)? However, the MS uses several other dates, including one six months after the specified date, which may over-state actual compliance? As Table S3 has the data, could you be more specific? 2. Could you outline some specific policies in the conclusions? I appreciate the word limit, but outlining a priority would provide clearer guidance. Summary 3. The final two bullet points seemed more like assumptions regarding the data rather than strengths or limitations of the study itself. I wondered if these bullet points would be better located in the methods section (where you do mention them) and if you could amplify point 2, where you rather undersell the study's strengths? Background 4. I wondered if the helpful distinctions made in Table 3 might be useful earlier in the Background and if you could use the terms in this table consistently throughout the text? For example, I think you mean brand names in para 2, p6 but I did not know whether you meant brand or variant (descriptor) names in para 3, where you refer to 'evocative names'? If the legislation prohibits evocative names, how could tobacco companies introduce words such as 'bright' and 'legendary' (p21)? 5. Box 1 is very helpful, but would it be possible to align the attributes so readers can more easily compare attributes across the UK legislation and EU TPD? Methods 6. I felt unsure what the cover wrap noted on p10 referenced – cigar and pipe tobacco or multiple cigarette packs? I assume the latter (given later comments) – could you consider two sentences to avoid confusion between product categories here? 7. The methods section clearly outlines a thoughtful and very comprehensive study. Congratulations on designing (and completing!) such an ambitious piece of work. 8. Could you comment more on Table 3 (in the Results) and the reduction in brand families and variants, but increase in SKUs? 9. I understand you have included details of all components of this ambitious study, but wondered whether you needed to outline the focus groups, given you do not cite any data from these? Tightening the methods section so it refers only to the data reported would streamline the MS a little. Results 10. Again, please consider providing data on compliance on the actual transition date. 11. Could you clarify the marketing to retailers (p19) – did tobacco companies use larger incentives (value increased)? Or do you mean more tobacco companies used incentives (strategy became more mainstream)? Or they offered incentives to more retailers (expanded use of the strategy)? 12. I was interested to see yellow was used to indicate 'smooth' on RYO tobacco. Some of my work with smokers has found they dislike yellow, which reminds them of nicotine stains and the 'dirtiness' of smoking. (This comment isn't a request for further comment, just an observation.) 13. How did you define 'the purpose of practical information' (p20)? 14. Table 4 suggests capsules remained permissible – I wasn't sure
--	---

how that sat with a ban on filter innovation technologies (p6)? Could you also give an example of the mixed and miscellaneous discretionary actions (Table 4, p21)?

15. If it is possible to give an example of materials provided to retailers, I think these would be of wide interest (perhaps a supplementary file)?

16. It is interesting to see the increase in RYO variants, which other work suggests is a lower-cost alternative to quitting.[8-10]

17. It was very interesting to see that 'premium' features appeared in value brands; is there any evidence whether UK smokers' quality perceptions remained similar post plain packaging? In Australia, some smokers considered the quality of tobacco to have declined post PP.[11]

Discussion

18. In places, the discussion seemed to repeat the results and I wondered if you could focus more on the policy implications and emphasise these more?

19. Occasionally, the tone became somewhat outraged (p28, l55ff.) I understand why, but wonder whether a more disinterested voice could prove as effective?

20. The multiple data sources represent a real strength of your work – could you trace the pathway from industry advertising, through pack observations and sales data to highlight how you used these different sources?

Supplementary Files

21. Table S2 was extremely interesting – I wondered whether this table justified a separate paper as it contains rich and very important data on industry activities? You may already have something planned so this comment is not a suggested change, rather a reflection that these data seem more important than a supplementary file!

Pedantic points

22. Please consider using 'advertisement/s' rather than 'advert/s'.

23. Please check use of commas.

24. Please consider replacing 'Furthermore' p6 l57 with 'However'?

25. Please use words when referring to numbers under 10.

Overall, a very ambitious and important study; thank you for the opportunity to comment and best wishes with future work.

Possible References

[1] Haggart K, Hoek J, Blank M. Market performance of capsule cigarettes in New Zealand. *Nicotine & Tobacco Research* 2018.

[2] Moodie C, Ford A, Dobbie F, et al. The power of product innovation: Smokers' perceptions of capsule cigarettes. *Nicotine & Tobacco Research* 2017.

[3] Hoek J, Gendall P, Eckert C, et al. Young adult susceptible non-smokers' and smokers' responses to capsule cigarettes. *Tobacco control* 2018;tobaccocontrol-2018-054470.

[4] Hoek J, Gendall P, Eckert C, et al. Effects of brand variants on smokers' choice behaviours and risk perceptions. *Tob Control* 2015;25(2):160-165.

[5] Borland R, Savvas S. The effects of variant descriptors on the potential effectiveness of plain packaging. *Tobacco control* 2014;23(1):58-63.

[6] Greenland SJ. Cigarette brand variant portfolio strategy and the

	use of colour in a darkening market. Tobacco control 2013:tobaccocontrol-2013-051055. [7] Hoek J, Gendall P. Policy options for extending standardized tobacco packaging. Bulletin of the World Health Organization 2017;95(10):726. [8] Hoek J, Ferguson S, Court E, et al. Qualitative exploration of young adult RYO smokers' practices. Tob Control 2016;26(5):563-568. [9] Bayly M, Scollo MM, Wakefield MA. Who uses rollies? Trends in product offerings, price and use of roll-your-own tobacco in Australia. Tobacco control 2018:tobaccocontrol-2018-054334. [10] Curti D, Shang C, Ridgeway W, et al. The use of legal, illegal and roll-your-own cigarettes to increasing tobacco excise taxes and comprehensive tobacco control policies: findings from the ITC Uruguay survey. Tobacco control 2015:tobaccocontrol-2014-051890. [11] Wakefield M, Hayes L, Durkin S, et al. Introduction effects of the Australian plain packaging policy on adult smokers: a cross-sectional study. BMJ open 2013;3(7).
--	--

REVIEWER	Ilze Bogdanovica University of Nottingham, UK One of the authors is listed as my mentor on my current fellowship.
REVIEW RETURNED	31-Jan-2019

GENERAL COMMENTS	This is a well written and important article highlighting opportunities to improve tobacco control policy. I have only minor comments:  1) Authors refer to 'switching names and sizes'. Please explain how was that switching, particularly sizes for cigarettes measured. 2) Background: "The current legislation does not permits the use of colour descriptors, bevelled edges on packs and cigarette filter technology innovations"- this should be clarified as later in the text authors state that colour descriptors are used. 3) Background: Authors suggest that previous research has only explored industry strategies up to two months post-legislation and that added value of the current study is that it looks at 7 months post-legislation. As there is such distinction it should be reported in results/ discussion whether findings from this study are different from those previously reported and whether there have been new or more intense developments in more recent months studied. 4) Methods: Authors suggest that one of the datasources used was purchases of eight top-selling tobacco brands. It should be explained how these were selected. 5) Methods: Authors need to specify what methods were used to analyse Nielsen data- was it comparison of monthly data or comparison between two specified data points etc. If possible please explain how did you use sales data to examine compliance with the removal of branding and pricemarking. 6) Methods: In section on name changes please explain how you have defined price segments and why July 2014 was chosen as a starting point. 7) Results: Please explain why you have used Nov 2017 for Nielsen data as a comparison point either in methods or results section. As the background section suggested that this study provides evidence on data covering 7 months post-legislation there should be consistent reporting. Other sources used for the study include data
---

	up until Dec 2017 and Nielsen is the only exception when data until Nov 2017 are used although more recent Nielsen data exist. 8) Results: Strategy 2 explains that there were a lot of name changes happening over the time period considered and only a proportion of those were attributable to plain packaging legislation requirements. It would be important to a) describe when these changes were observed (was it before or during the implementation phase? or once products were introduced in plain packs?) and b) specify what criteria were used to decide whether the name changed was associated with the legislation 9) Results: In the section describing strategy 3 authors suggest that 68% of new variants recorded by Nielsen were in FM, but Table 5 refers to November 2018. This should be explained or corrected. 10) Results: Currently authors have considered brand variants and looked at changes in the number of brand variants. It would be worth adding information on what happens to SKU- this would be particularly important as somehow it needs to be explained what happens to cigarettes sold in 10-19 packs and whether any of the changes observed in brand variant name were also related to changes in pack size. 11) Results: For strategy 4 authors suggest that following full implementation sales of larger packs increased from 7 to 11 million packs. It should be clarified what time period this comparison is referring to, and whether the trend changed with the implementation of the legislation or remained the same. 12) Discussion: Page 28 Line 19 "... introduced smaller pack sizes to enable more affordable offerings...". Results section does not explain what happened to smaller pack sizes so if this statement is based on current study it should be included in the results or alternatively a relevant reference should be provided. 13) Discussion: Authors have reported that brand variant names were changed to include colours and it has been interpreted that this leads to misperceptions about harms. It would be useful if authors could cite existing evidence on how these colours are perceived by smokers if such evidence exist. 14) Discussion: Page 30 Line 40 "...and an evaluation of the introduction of standardised packaging in Australia"- please provide relevant references for this statement.
--	---

REVIEWER	Katherine Clegg Smith Johns Hopkins Bloomberg School of Public Health, USA
REVIEW RETURNED	07-Feb-2019

GENERAL COMMENTS	This prospective study of tobacco industry actions during the implementation of plain packaging policy provides important insight for countries considering such actions now or in the near future. This work is an important contribution to the global tobacco control community. In the objectives for the study, it seems to me to that it would be important to also state that the study is particularly focused on the 'sell-through' period, and tobacco industry actions during this time that may delay or compromise the impact of plain packaging policies in terms of protection against misleading marketing via the pack. While the multiple data sources do provide a valuable comprehensive view on industry activities during the implementation/transition phase, they did make for some challenges for the reader in terms of following the big points being made, and assessing how the data generated contributed to the findings and
--

	conclusions. There is just a whole lot here and I struggled to hold on to it all. I would welcome any simplification/additional organization that the authors could undertake with a goal of simplification of data source/study findings/conclusions drawn. In the data sources section, the specific analyses/questions to which each source would be applied could be made more explicit. Perhaps include the data source in brackets for each of the elements of the strategies outlined in the results section? It is included for some points, but not for others. For table 2, it is important to explain sampling for commercial literature – units examined within the publications (examined/available). I would find it helpful if the legends (or titles) on the tables included the data source. Implications of new brand variants and name changes at the lower end of the market could be discussed more in the discussion section. Minor:  • Page 6, line 52, should this read that beveled edges, color descriptors etc are still allowed (“current legislation does not permits the use” doesn’t make sense and seems counter to the point being made). • Page 7 lines 32-34. It seems that the analysis goes beyond considerations of compliance and circumvention of the law and also examines how the industry attempts to reduce the law’s immediate impact by actions during the ‘sell through’ period. • I found the set up of Box 1 to be very confusing. Do the columns and rows correspond? I would suggest restructuring – and possibly creating 2 boxes. • A few lines walking through the information in table 1 would be useful. • Check that data are plural throughout.
--	--

VERSION 1 – AUTHOR RESPONSE

Reviewer: 1

Reviewer Name: Janet Hoek

Institution and Country: Departments of Public Health and Marketing, University of Otago, New Zealand Please state any competing interests or state ‘None declared’: I have met Anna Gilmore, who hosted me when I visited the University of Bath in 2015.

I enjoyed reading this thoughtful and very comprehensive study; it makes a new contribution by drawing together different data sources and triangulating the evidence from these. I have a few suggestions that I hope will be useful, and have outlined these below.

Thank you very much for the time and effort you have put into reviewing our manuscript and for your comments.

General Comments

I thought the MS could have drawn more on the international literature; the Scollo references from Australia are very important and well used, but there are also other studies documenting product innovation and how this accelerated as plain packaging was announced.[1] Earlier work also discusses how innovation could threaten plain packaging’s effect on deterring initiation;[2, 3] outlines the effects of variant names [4, 5] and colours, [6] and overviews policy loopholes and how these might be addressed.[7] These are just a few of the papers I know about that could illustrate how UK

evidence both reflects and is supported by international data. Given how comprehensive the data sources are, I hope you will see value in expanding the literature review.

Thank you very much. We have taken your advice and expanded the literature review and references cited in the background section.

In places, the MS would benefit from some editing and careful proofreading. For example, the references have numerous errors and I'm not sure how easy it would be for someone not involved in the project to locate some items. EndNote does strange things to some references (see refs 1,2, 30, for example) and all the capitals seem to have disappeared (e.g. from ITC, UK etc.). Shortening sentences, reducing prepositional phrases, employing the active voice, and using more precise language could make the MS easier to read. Please also check your use of commas.

Thank you. We have attempted to address the errors in the referencing. We have shortened sentences, checked my use of commas and attempted to use the active voice and therefore we hope the manuscript is now easier to read.

Some specific suggestions are:

Abstract

1. Could you use a single date for assessing compliance? I would have thought the best date was the actual date of transition (May 20, 2017)? However, the MS uses several other dates, including one six months after the specified date, which may over-state actual compliance? As Table S3 has the data, could you be more specific?

Thank you. It was not possible to assess compliance using the Nielsen dataset on 20 May 2017 as the monthly data included all sales between 1 and 19th May too. The first month post legislation is the closest we could get so we've included data from June 2017. We have made this clearer in the first paragraph of the results and added the one-month compliance data to the results section of the abstract. In addition to compliance we wanted to know how tobacco companies adapted to the regulations and what, if any, changes they made to their products and the way that they promoted any changes in the retail trade press not only by the actual date of transition but also post-legislation. We suspected that tobacco companies continued to make adaptations to their products and their marketing to retailers up to the present day and so when it came to analysing the data for this manuscript we wanted to use everything we had to provide as comprehensive a view as possible. Given that we had so many different datasets (with differing start and end dates), it means that the methods and results were difficult to follow. We have now included a diagram in the methods detailing the start and end dates of each dataset and included a sentence explaining why we haven't selected the same start and end dates for each dataset.

2. Could you outline some specific policies in the conclusions? I appreciate the word limit, but outlining a priority would provide clearer guidance.

We have specified standardised packaging and tried to provide clearer guidance in line with your suggestion.

Summary

3. The final two bullet points seemed more like assumptions regarding the data rather than strengths or limitations of the study itself. I wondered if these bullet points would be better located in the methods section (where you do mention them) and if you could amplify point 2, where you rather undersell the study's strengths?

We have added more emphasis to point 2 and included a more comprehensive description about why the final two points are limitations of the study. We hope it is now clearer that these are limitations and do belong in the strengths and limitations summary of the paper.

Background

4. I wondered if the helpful distinctions made in Table 3 might be useful earlier in the Background and if you could use the terms in this table consistently throughout the text? For example, I think you

mean brand names in para 2, p6 but I did not know whether you meant brand or variant (descriptor) names in para 3, where you refer to 'evocative names'? If the legislation prohibits evocative names, how could tobacco companies introduce words such as 'bright' and 'legendary' (p21)?

We have tried to make the distinctions clearer in the Background and Table 3 is now Table 1 so that the definitions are cited earlier in the manuscript. We agree that bright and legendary are evocative, however, having reviewed the terminology of both the EU and UK legislation, my description of evocative is not entirely accurate – the prohibition refers to names that may create an erroneous perception about the characteristics, health effects, risks or emissions of tobacco. Therefore, we think strictly, this is a circumvention but not a breach of the regulations. We have changed the wording in the introduction to more accurately describe the regulations.

5. Box 1 is very helpful, but would it be possible to align the attributes so readers can more easily compare attributes across the UK legislation and EU TPD?

Box 1 has been redesigned to list the EU legislation first and the UK legislation second. We think this makes it clearer. It's not really possible to align them as most deal with slightly different aspects of packaging.

Methods

6. I felt unsure what the cover wrap noted on p10 referenced – cigar and pipe tobacco or multiple cigarette packs? I assume the latter (given later comments) – could you consider two sentences to avoid confusion between product categories here?

Yes, we meant the latter. We've added some words to the paragraph. Thank you.

7. The methods section clearly outlines a thoughtful and very comprehensive study. Congratulations on designing (and completing!) such an ambitious piece of work.

Thank you so much. We appreciate you taking the time to tell us this.

8. Could you comment more on Table 3 (in the Results) and the reduction in brand families and variants, but increase in SKUs?

We included the following paragraph:

“Overall, tobacco companies decreased the number of brand families and brand variants between July 2014 and November 2017 as they were consolidating their portfolios and focusing on the brands which could offer growth. The increase in SKU between 2014 and 2017 reflects (a) the pre-standardised packs increase in small pack sizes, innovations such as flavour capsules (b) the addition of standardised packs FM SKU and 30g RYO SKU and (c) that many SKU in branded packs remained on the market in November 2017 but selling only tiny volumes.”

9. I understand you have included details of all components of this ambitious study, but wondered whether you needed to outline the focus groups, given you do not cite any data from these? Tightening the methods section so it refers only to the data reported would streamline the MS a little.

We have deleted the focus groups from the manuscript – this was an oversight from an earlier draft of the MS. Thanks so much.

Results

10. Again, please consider providing data on compliance on the actual transition date.

As per our earlier comment – given the limitations of the May 2017 data from Nielsen, we utilised the data for one month post-implementation in June 2017 which is included under Strategy 1 of the results.

11. Could you clarify the marketing to retailers (p19) – did tobacco companies use larger incentives (value increased)? Or do you mean more tobacco companies used incentives (strategy became more mainstream)? Or they offered incentives to more retailers (expanded use of the strategy)?

We have included the following paragraph in the results:

“Tobacco companies offered incentives and promotions on branded packs to retailers to encourage sales and loyalty to their products, examples included cash rewards and loyalty points for selling particular brands. By early 2017 manufacturers were described as being in a price-war.(17, 18)”

12. I was interested to see yellow was used to indicate ‘smooth’ on RYO tobacco. Some of my work with smokers has found they dislike yellow, which reminds them of nicotine stains and the ‘dirtiness’ of smoking. (This comment isn’t a request for further comment, just an observation.)

Interesting. We have not uncovered evidence on why they used different colours to represent smooth. In the case of yellow, the main brand where this convention was used was ‘Golden Virginia’ so it may be related to ‘Golden’ but we have no documentary evidence to support this link.

13. How did you define ‘the purpose of practical information’ (p20)?

We have changed the wording to explain that there were 23 variants that did not simply add a colour adjective but added other novel elements – we now refer to these as novel elements

14. Table 4 suggests capsules remained permissible – I wasn’t sure how that sat with a ban on filter innovation technologies (p6)? Could you also give an example of the mixed and miscellaneous discretionary actions (Table 4, p21)?

Apologies, there was a typo in the background section, there are regulations on how the filter should look (e.g. like cork, no white filters), but there are no regulations on innovating how the filter works or what it is composed of. So there was NO ban on filter innovation technologies.

We’ve also added the following to Table 4 to provide examples of the mixed and miscellaneous findings.

Mixed: e.g. colour descriptor added and a size was removed

Miscellaneous e.g. some names had a location added or removed e.g. a Rothmans brand family was created called Rothman’s of London which later became Rothman’s Blue

15. If it is possible to give an example of materials provided to retailers, I think these would be of wide interest (perhaps a supplementary file)?

Pictures of some of the materials are available in advertorials in the trade press - we have referenced an example. Sadly, we are not permitted to republish the advertorial due to copyright restrictions. We did try to get physical copies of the actual materials from retailers that we could have taken pictures of and published the images but sadly our requests were unsuccessful.

16. It is interesting to see the increase in RYO variants, which other work suggests is a lower-cost alternative to quitting.[8-10]

Yes, thank you. We have now included this excellent observation in the discussion section.

17. It was very interesting to see that ‘premium’ features appeared in value brands; is there any evidence whether UK smokers’ quality perceptions remained similar post plain packaging? In Australia, some smokers considered the quality of tobacco to have declined post PP.[11]

We have checked the UK literature for this and no studies have been conducted here to quality perceptions of tobacco products post-implementation.

Discussion

18. In places, the discussion seemed to repeat the results and I wondered if you could focus more on the policy implications and emphasise these more?

Thank you. We were attempting to highlight the four main circumventions we found and then put the policy implications at the end of each circumvention and go one step further and make a recommendation based on the policy implications. We have tried to redress the balance of the discussion to reduce repetition.

19. Occasionally, the tone became somewhat outraged (p28, l55ff.) I understand why, but wonder whether a more disinterested voice could prove as effective?

Thank you. We have attempted to amend the tone.

20. The multiple data sources represent a real strength of your work – could you trace the pathway from industry advertising, through pack observations and sales data to highlight how you used these different sources?

Unfortunately this is not possible due to the timings of data collection for each dataset. We would risk overstepping the constraints of the data. We might be able to do it for one brand where each of the datasets holds information for that particular brand in a chronological manner but this is of limited utility for just one brand.

Supplementary Files

21. Table S2 was extremely interesting – I wondered whether this table justified a separate paper as it contains rich and very important data on industry activities? You may already have something planned so this comment is not a suggested change, rather a reflection that these data seem more important than a supplementary file!

Thank you. We hadn't previously considered and it is now under discussion.

Pedantic points

22. Please consider using 'advertisement/s' rather than 'advert/s'.

23. Please check use of commas.

24. Please consider replacing 'Furthermore' p6 l57 with 'However'?

25. Please use words when referring to numbers under 10.

Done. Thank you.

Overall, a very ambitious and important study; thank you for the opportunity to comment and best wishes with future work.

Thank you again for your time and a through and extremely helpful review.

Possible References

[1] Haggart K, Hoek J, Blank M. Market performance of capsule cigarettes in New Zealand. Nicotine & Tobacco Research 2018.

[2] Moodie C, Ford A, Dobbie F, et al. The power of product innovation: Smokers' perceptions of capsule cigarettes. Nicotine & Tobacco Research 2017.

[3] Hoek J, Gendall P, Eckert C, et al. Young adult susceptible non-smokers' and smokers' responses to capsule cigarettes. Tobacco control 2018:tobaccocontrol-2018-054470.

[4] Hoek J, Gendall P, Eckert C, et al. Effects of brand variants on smokers' choice behaviours and risk perceptions. Tob Control 2015;25(2):160-165.

- [5] Borland R, Savvas S. The effects of variant descriptors on the potential effectiveness of plain packaging. *Tobacco control* 2014;23(1):58-63.
- [6] Greenland SJ. Cigarette brand variant portfolio strategy and the use of colour in a darkening market. *Tobacco control* 2013:tobaccocontrol-2013-051055.
- [7] Hoek J, Gendall P. Policy options for extending standardized tobacco packaging. *Bulletin of the World Health Organization* 2017;95(10):726.
- [8] Hoek J, Ferguson S, Court E, et al. Qualitative exploration of young adult RYO smokers' practices. *Tob Control* 2016;26(5):563-568.
- [9] Bayly M, Scollo MM, Wakefield MA. Who uses rollies? Trends in product offerings, price and use of roll-your-own tobacco in Australia. *Tobacco control* 2018:tobaccocontrol-2018-054334.
- [10] Curti D, Shang C, Ridgeway W, et al. The use of legal, illegal and roll-your-own cigarettes to increasing tobacco excise taxes and comprehensive tobacco control policies: findings from the ITC Uruguay survey. *Tobacco control* 2015:tobaccocontrol-2014-051890.
- [11] Wakefield M, Hayes L, Durkin S, et al. Introduction effects of the Australian plain packaging policy on adult smokers: a cross-sectional study. *BMJ open* 2013;3(7).

Reviewer: 2

Reviewer Name: Ilze Bogdanovica

Institution and Country: University of Nottingham, UK Please state any competing interests or state 'None declared': One of the authors is listed as my mentor on my current fellowship.

This is a well written and important article highlighting opportunities to improve tobacco control policy.

I have only minor comments:

1) Authors refer to 'switching names and sizes'. Please explain how was that switching, particularly sizes for cigarettes measured.

We've changed this to read "name changes and pack sizes in the Nielsen data were similar for..."

2) Background: "The current legislation does not permits the use of colour descriptors, bevelled edges on packs and cigarette filter technology innovations"- this should be clarified as later in the text authors state that colour descriptors are used.

Apologies, this is a typo in the background it should say "The current legislation permits..." It has been amended now. Thank you.

3) Background: Authors suggest that previous research has only explored industry strategies up to two months post-legislation and that added value of the current study is that it looks at 7 months post-legislation. As there is such distinction it should be reported in results/ discussion whether findings from this study are different from those previously reported and whether there have been new or more intense developments in more recent months studied.

Good point. We have added a couple of sentences to the discussion to reflect this.

4) Methods: Authors suggest that one of the datasources used was purchases of eight top-selling tobacco brands. It should be explained how these were selected.

Top-selling brands in 5 different price segments for cigarettes, 2 for RYO were selected. We also included the top selling BAT brand in the UK as a BAT brand was not a top-seller in any of the selected price segments. I've included this information in the methods section of the MS.

5) Methods: Authors need to specify what methods were used to analyse Nielsen data- was it comparison of monthly data or comparison between two specified data points etc.

Thank you. We've added to the methods section that this is a comparison of monthly sales data.

If possible please explain how did you use sales data to examine compliance with the removal of branding and pricemarking.

This information is included under the compliance section in the methods.

6) Methods: In section on name changes please explain how you have defined price segments and why July 2014 was chosen as a starting point.

We've added to the methods on pack purchasing that price segments were defined from commercial literature and Nielsen tobacco price data between 2008 and early 2016. We've added this information here as it's the first place in the MS that price segments are introduced.

In the Name changes section of the methods we've clarified that July 2014 was used as a starting point for this analysis as we wanted a time before standardised packaging was passed by the Government in January 2015.

7) Results: Please explain why you have used Nov 2017 for Nielsen data as a comparison point either in methods or results section. As the background section suggested that this study provides evidence on data covering 7 months post-legislation there should be consistent reporting. Other sources used for the study include data up until Dec 2017 and Nielsen is the only exception when data until Nov 2017 are used although more recent Nielsen data exist.

Nov 2017 was the latest available Nielsen data at the time of analysis.

8) Results: Strategy 2 explains that there were a lot of name changes happening over the time period considered and only a proportion of those were attributable to plain packaging legislation requirements. It would be important to a) describe when these changes were observed (was it before or during the implementation phase? or once products were introduced in plain packs?) and b) specify what criteria were used to decide whether the name changed was associated with the legislation

In preliminary analysis there were no clear patterns over time. This was not surprising because there was a year-long implementation period. The ones assumed to be due to standardised packs were changes that were mentioned in the legislation e.g. removal of flavours. We deemed it more important to assess the changes as they endured and whether these changes were just because of legislative requirements or because companies needed to maintain, or create a new, identity for a brand family/variant.

9) Results: In the section describing strategy 3 authors suggest that 68% of new variants recorded by Nielsen were in FM, but Table 5 refers to November 2018. This should be explained or corrected.

Apologies in the headers in table 5 for name changes and new variants the dates should be (July 2014 to Nov 2018)

10) Results: Currently authors have considered brand variants and looked at changes in the number of brand variants. It would be worth adding information on what happens to SKU- this would be particularly important as somehow it needs to be explained what happens to cigarettes sold in 10-19 packs

This has largely been covered in Breton MO, Britton J, Huang Y, Bogdanovica I. Cigarette brand diversity and price changes during the implementation of plain packaging in the United Kingdom. *Addiction*. 2018;113(10):1883-94.

However, we have now added our observation supported by Table 3 that despite the decrease in brand families and variants overall there has been an increase in SKU.

We added the following paragraph of explanation:

Overall, tobacco companies decreased the number of brand families and brand variants between July 2014 and November 2017 as they were consolidating their portfolios and focusing on the brands

which could offer growth. The increase in SKU between 2014 and 2017 reflects (a) the pre-standardised packs increase in small pack sizes and innovations such as flavours or capsules (b) the addition of standardised packs FM SKU and 30g RYO SKU and (c) that many SKU in branded packs remained on the market in November 2017 but selling only tiny volumes.

and whether any of the changes observed in brand variant name were also related to changes in pack size.

We looked at this in preliminary analysis and this was not a big issue: there was only one brand which had two brand variants which were differentiated by number of sticks in the pack. We wanted to stick closely to our main arguments and we worry this would be extraneous and not part of the bigger strategies employed by TTCs.

11) Results: For strategy 4 authors suggest that following full implementation sales of larger packs increased from 7 to 11 million packs. It should be clarified what time period this comparison is referring to, and whether the trend changed with the implementation of the legislation or remained the same.

We have changed the second sentence in this section to:

“During the sell-through (May 2016 - May 2017) sales of packs with 23 and 24 sticks increased from 7 to 18 million sticks. Sales peaked at 25 million sticks in August 2017 and declined to 21 million sticks in November 2017.

12) Discussion: Page 28 Line 19 “... introduced smaller pack sizes to enable more affordable offerings...”. Results section does not explain what happened to smaller pack sizes so if this statement is based on current study it should be included in the results or alternatively a relevant reference should be provided.

This point refers to the introduction of smaller pack sizes prior to the full implementation of the legislation. It was in the Nielsen data and also evident in the trade press. We've attempted to make this clearer in the results.

13) Discussion: Authors have reported that brand variant names were changed to include colours and it has been interpreted that this leads to misperceptions about harms. It would be useful if authors could cite existing evidence on how these colours are perceived by smokers if such evidence exist.

References 32 and 33 discuss this issue:

32. Mutti S, Hammond D, Borland R, Cummings MK, O'Connor RJ, Fong GT. Beyond light and mild: Cigarette brand descriptors and perceptions of risk in the international tobacco control (itc) four country survey. *Addiction*. 2011;106(6):1166-1175.

33. Bansal-Travers M, O'Connor RJ, Fix BV, Cummings MK. What do cigarette pack colors communicate to smokers in the u.S.? *American Journal of Preventive Medicine*. 2011;40(6):683-89.

14) Discussion: Page 30 Line 40 “...and an evaluation of the introduction of standardised packaging in Australia”- please provide relevant references for this statement.

Apologies and thank you for spotting this omission. The studies we were referring to were:

Scollo M, Occleston J, Bayly M, Lindorff K, Wakefield M. Tobacco product developments coinciding with the implementation of plain packaging in Australia. *Tobacco Control*. 2014 April 30, 2014.

Scollo M, Bayly M, White S, Lindorff K, Wakefield M. Tobacco product developments in the Australian market in the 4 years following plain packaging. *Tobacco Control*. 2017.

Reviewer: 3

Reviewer Name: Katherine Clegg Smith

Institution and Country: Johns Hopkins Bloomberg School of Public Health, USA Please state any competing interests or state 'None declared': None declared

This prospective study of tobacco industry actions during the implementation of plain packaging policy provides important insight for countries considering such actions now or in the near future. This work is an important contribution to the global tobacco control community.

Thank you very much for sharing our view point that this manuscript has an important contribution to make to global tobacco control.

In the objectives for the study, it seems to me to that it would be important to also state that the study is particularly focused on the 'sell-through' period, and tobacco industry actions during this time that may delay or compromise the impact of plain packaging policies in terms of protection against misleading marketing via the pack.

Thank you for your observation. Indeed part of the study is focused on the sell-through period, however we do also focus on a considerable period of time post-legislation. We have included a diagram in the methods to better illustrate the time periods covered by the datasets in the study.

While the multiple data sources do provide a valuable comprehensive view on industry activities during the implementation/transition phase, they did make for some challenges for the reader in terms of following the big points being made, and assessing how the data generated contributed to the findings and conclusions. There is just a whole lot here and I struggled to hold on to it all. I would welcome any simplification/additional organization that the authors could undertake with a goal of simplification of data source/study findings/conclusions drawn. In the data sources section, the specific analyses/questions to which each source would be applied could be made more explicit. Perhaps include the data source in brackets for each of the elements of the strategies outlined in the results section? It is included for some points, but not for others.

Thank you, we agree. We hope that the diagram included in the methods section has helped clarify the datasets, the time periods and the key questions asked of each dataset. We considered your suggestion of adding which dataset was being used to answer each question in brackets but opted instead to try and clarify the data source at the outset of each sentence in the results.

For table 2, it is important to explain sampling for commercial literature – units examined within the publications (examined/available). I would find it helpful if the legends (or titles) on the tables included the data source.

We have changed the table headings for the purpose of clarity

Source	Search terms (depends on search procedure of website)	Articles/ reportsfound	Articles/ reports included in final analysis
---------------	--	-------------------------------	---

Implications of new brand variants and name changes at the lower end of the market could be discussed more in the discussion section.

We've expanded this section a little.

Minor:

- Page 6, line 52, should this read that beveled edges, color descriptors etc are still allowed ("current legislation does not permits the use" doesn't make sense and seems counter to the point being made).

Yes. Thank you. The sentence in the background section included a typo which confused everything. Amended now.

- Page 7 lines 32-34. It seems that the analysis goes beyond considerations of compliance and circumvention of the law and also examines how the industry attempts to reduce the law's immediate impact by actions during the 'sell through' period.

Yes, this is a good point, one could argue that the attempts to reduce the law's immediate impact by actions during the sell-through is a form of circumvention and I've amended the MS to better reflect this.

- I found the set up of Box 1 to be very confusing. Do the columns and rows correspond? I would suggest restructuring – and possibly creating 2 boxes.

Agreed. The Box has been reformatted to present the EU regulations first and the UK regulations second.

- A few lines walking through the information in table 1 would be useful.

We've added a couple of lines about the price segmentation of the data in table 1.

- Check that data are plural throughout.

Done. Thank you.